# Therapeutic Peptide RF16 Derived from CXCL8 Inhibits MDA-MB-231 Cell Invasion and Metastasis

**DOI:** 10.3390/ijms241814029

**Published:** 2023-09-13

**Authors:** Chun-Ming Chang, Chun-Chun Chang, Ho Yin Pekkle Lam, Shih-Yi Peng, Yi-Hsuan Lai, Bi-Da Hsiang, Yu-Yi Liao, Hao-Jen Hsu, Shinn-Jong Jiang

**Affiliations:** 1Department of General Surgery, Hualien Tzu Chi Hospital, Buddhist Tzu Chi Medical Foundation, Hualien 97004, Taiwan; ccmjim1008@gmail.com; 2Institute of Medical Sciences, Tzu Chi University, Hualien 97004, Taiwan; 3Department of Laboratory Medicine, Hualien Tzu Chi Hospital, Hualien 97004, Taiwan; chunchun@mail.tcu.edu.tw; 4Department of Laboratory Medicine and Biotechnology, College of Medicine, Tzu Chi University, Hualien 97004, Taiwan; 5Department of Biochemistry, School of Medicine, Tzu Chi University, Hualien 97004, Taiwan; pekklelavabo@mail.tcu.edu.tw (H.Y.P.L.); pengsy@mail.tcu.edu.tw (S.-Y.P.); jokerlai0383315@gmail.com (Y.-H.L.); 6Master Program in Biomedical Sciences, School of Medicine, Tzu Chi University, Hualien 97004, Taiwan; 111333106@gms.tcu.edu.tw; 7Department of Molecular Biology and Human Genetics, School of Medicine, Tzu Chi University, Hualien 97004, Taiwan; 108712125@gms.tcu.edu.tw; 8Department of Biomedical Sciences and Engineering, College of Medicine, Tzu Chi University, Hualien 97004, Taiwan; hjhsu32@mail.tcu.edu.tw

**Keywords:** breast cancer, IL-8, peptide drugs, cell proliferation, invasiveness, metastasis

## Abstract

Interleukin (IL)-8 plays a vital role in regulating inflammation and breast cancer formation by activating CXCR1/2. We previously designed an antagonist peptide, (RF16), to inhibits the activation of downstream signaling pathways by competing with IL-8 in binding to CXCR1/2, thereby inhibiting IL-8-induced chemoattractant monocyte binding. To evaluate the effect of the RF16 peptide on breast cancer progression, triple-negative MDA-MB-231 and ER-positive MCF-7 breast cancer cells were used to investigate whether RF16 can inhibit the IL-8-induced breast cancer metastasis. Using growth, proliferation, and invasiveness assays, the results revealed that RF16 reduced cell proliferation, migration, and invasiveness in MDA-MB-231 cells. The RF16 peptide also regulated the protein and mRNA expressions of epithelial–mesenchymal transition (EMT) markers in IL-8-stimulated MDA-MB-231 cells. It also inhibited downstream IL-8 signaling and the IL-8-induced inflammatory response via the mitogen-activated protein kinase (MAPK) and Phosphoinositide 3-kinase (PI3K) pathways. In the xenograft tumor mouse model, RF16 synergistically reinforces the antitumor efficacy of docetaxel by improving mouse survival and retarding tumor growth. Our results indicate that RF16 significantly inhibited IL-8-stimulated cell growth, migration, and invasion in MDA-MB-231 breast cancer cells by blocking the activation of p38 and AKT cascades. It indicated that the RF16 peptide may serve as a new supplementary drug for breast cancer.

## 1. Introduction

Breast cancer is one of the most common malignant tumors in women, and its incidence is increasing every year [1]. Age, ethnicity, and body mass index are key determinants of cancer progression [2]. Despite advancements in the diagnosis and management of malignant and local tumors [3,4], numerous patients develop this recurrent and metastatic disease [5]. Breast cancer is primarily managed with surgery and chemotherapy. However, in vitro and in vivo studies have declared that breast CSCs (cancer stem cells) are counteractive to both chemotherapy and radiation [6,7]. Clinical studies have also confirmed that after neoadjuvant chemotherapy, the percentage of breast CSCs is enhanced [8,9,10]. This indicates that recurrence is a common phenomenon in breast cancer, especially the triple-negative breast cancers. Tumor cells invasion and metastasis often result in poor prognosis. Penetration into the extracellular basement membrane is a prerequisite for cancer cell metastasis. Accordingly, many factors including certain proteases play a vital role in it.

With the development of new molecular biology technologies, an increasing number of studies are being conducted to elucidate the mechanisms related to breast cancer, particularly those related to growth factors and cytokines that regulate disease development and tumor microenvironment, thus promoting tumor growth and survival [2]. Interleukin-8 (IL-8; CXCL8) is a member of the chemokine CXC family and a key inflammatory chemokine. It induces macrophage and neutrophil infiltration. In some inflammatory diseases, such as cystic fibrosis and chronic obstructive pulmonary disease, the cells express large numbers of related chemokines, including IL-8 [11,12,13]. IL-8 mainly acts by activating a series of downstream reactions by binding to CXCR1 and CXCR2, and IL-8 has a relatively higher specific binding to CXCR1 [14,15]. These downstream signals can promote the angiogenesis of endothelial cells, the proliferation and survival of endothelial and cancer cells, the migration and epithelial–mesenchymal transition (EMT) of cancer cells, and neutrophil recruitment to tumor sites [16,17,18,19]. Accordingly, high expressions of IL-8 and CXCR1/2 are associated with many tumors, including endometrial cancer, pancreatic cancer, liver cancer, and lung cancer, and patients with high IL-8 levels have poor survival [20,21,22,23]. It also demonstrated that CXCR1 expression was increased in the breast CSC population [20]. Moreover, recombinant IL-8 adjunction enhanced the CSC population as well as increasing its proclivity for invasion [24]. Overexpression of the IL-8 receptor CXCR1/2 on cancer cells stimulates survival signals [20,25,26], such as PI3K/AKT pathway activation [27,28,29]. Tumor cells with acquired resistance exhibit constitutive AKT activation, and chemotherapeutic agents sometimes activate this survival mechanism during cancer treatment [30,31]. Another pathway that IL-8 activates is the MAPK/ERK signaling pathway in neutrophils and cancer cells [27,28,29,32]. During EMT, epithelial cells undergo a morphological change and lose cells adhesion via the apical–basal polarity [33]. As a result, they acquire high invasive and metastatic potential and can cause extracellular matrix (ECM) degradation, which is closely related to cancer development and progression [33]. If the tumor intracellular microenvironment plays a significant role in the modulation of CSC proliferation and survival, strategies to block these interactions denote a reasonable approach to target breast cancer cells.

We previously used in silico software to predict the binding between IL-8 and CXCR1/2 and designed antagonist peptides that may inhibit the binding of IL-8 and CXCR1/2 [34]. IL-8 induces mononuclear cell adhesion, and antagonist peptides reduce the number of IL-8-induced mononuclear spheres that cross the endothelial cell layer [34]. Ginestier et al. demonstrated that blocking the CXCR1 activation selectively reduced the breast CSC population in vitro and in immunocompromised xenograft models. The administration of repertaxin, a small-molecule CXCR1 inhibitor, alleviated tumor development and decreased the progression of breast cancer metastasis in NOD/SCID mice [20]. This suggests that strategies focused on intervening in the IL-8/CXCR1 axis may be capable of targeting CSCs and decreasing the growth of breast cancer, increasing the usefulness of present therapies.

In this study, we investigated the effect of one of these antagonist peptides, RF16, on breast cancer cell growth to identify a more effective treatment for breast cancer. We used two human breast cancer cell lines: low-metastatic estrogen receptor (ER)+ MCF-7 cells and highly metastatic ER− MDA-MB-231 cells to compare the efficacy of the RF16 peptide on breast cancer cells with different features. Our results suggest that RF16 blocks MDA-MB-231 cell invasion and metastasis by inhibiting cell proliferation, migration, and EMT. In the in vivo xenograft tumors model, we determined that RF16 synergistically increased the antitumor efficacy of docetaxel. It suggested that the RF16 peptide may provide us with a new supplementary drug for breast cancer therapy.

## 2. Results

### 2.1. RF16 Bound to the CXCR1/2 N-Terminal Region and CXCL8 Bound to CXCR2

We previously demonstrated that the peptide p_wt14 (8QCIKTYSKPFHPKF21) can bind to the N-terminus of CXCR1 in molecular modeling, SPR measurements and cellular experiments theoretically indicated that RF16 (6RCQCIKTYSKPFHPKF21), which is two residues more than p_wt14, can also bind to CXCR1 [34]. In this study, the binding of RF16 to the N-terminal region of CXCR2 was demonstrated resembling the complex structure of CXCL8¬–CXCR2 (Figure 1). The molecular docking of RF16 bound to CXCR1/2 indicated that electrostatic interactions dominated the RF16 docking to the N-terminus of CXCR1/2 (Figure 1). The positively charged residues of RF16 (R6, K11, K15, and K20) interacted with the negatively charged residues at the N-terminals of CXCR1 (D6, D11, D13, D14, D24, E25, and D26) and CXCR2 (D35, E40, E42, and E45). The detail interaction between RF16 with CXCR1 or CXCR2 could be seen in Appendix A.

### 2.2. RF16 Reduced IL-8-Stimulated Breast Cancer Cell Growth and Proliferation 

We evaluated the potential role of RF16–CXCL1/2 binding on the growth and proliferation of MCF-7 and MDA-MB-231 cells. CF25, a scramble peptide with 25 random amino acids, was used as a negative control. The results revealed that 100 μM RF16 only slightly decreased MCF-7 cell growth compared with the control cells at 48 and 72 h (Figure 2A), whereas RF16 dose-dependently inhibited MDA-MB-231 cell proliferation at 72 h (Figure 2B).

We then determined the cytotoxicity of RF16 in MCF-7 (Figure 2C) and MDA-MB-231 cells (Figure 2D). As indicated in Figure 2B,D, 0.001–100 μM RF16 did not induce cytotoxicity in either cell line, implying that the RF16-induced decrease in MDA-MB-231 cell proliferation was not due to cytotoxicity.

A colony formation assay was performed to visualize whether RF16, at noncytotoxic concentrations, can influence the colony formation of breast cancer cells. We observed that RF16 did not affect the colony-forming activity of MCF-7, whereas decreasing the IL-8 levels increased the size of the colonies of MDA-MB-231 cells (Figure 3A,B). These results indicate that RF16 significantly suppressed MDA-MB-231 cell proliferation, whereas CF25 had no effect on cancer cell growth or proliferation in either cell lines.

### 2.3. RF16 Suppressed IL-8-Activated Breast Cancer Cell Migration and Invasion

Because the RF16 peptide reduced breast cancer cell growth and proliferation, we next investigated whether RF16 could alter breast cancer cell migration and invasion. We assessed the migratory potential of MCF-7 and MDA-MB-231 cells treated with different concentrations of RF16 through a wound-healing assay. IL-8 treatment did not cause MCF-7 cells to migrate across the wound area compared with the untreated control at 24 h. The wound-healing rate in the RF16-treated MCF-7 cells did not differ between the control and IL-8-stimulated cells (Figure 4A). By contrast, the ability of IL-8-treated MDA-MB-231 cells to migrate was notably elevated compared with the control cells, whereas RF-16 treatment remarkably suppressed wound migration in a dose-dependent manner by approximately 60–90% compared with the control cells (Figure 4B). The CF25 peptide exhibited no inhibitory ability on breast cancer cells. Because the effect of IL-8 on MCF-7 was nonsignificant, the function of RF16 was focused on highly metastatic MDA-MB-231 cells in subsequent experiments. When the Matrigel invasion assay was conducted on MDA-MB-231 cells, IL-8 more than doubled invasiveness, and RF16 decreased the number of IL-8-upregulated invasive cells (Figure 5A). The results strongly suggest that RF16 reduced the IL-8-promoted migration and invasion of MDA-MB-231 cells.

To further investigate the invasive ability of breast tumor cells, proteins related to invasion were assayed in MDA-MB-231 cells. MMP expression was reduced in MDA-MB-231 cells treated with RF16 at the protein activity level, as detected using an MMP activity analysis (Figure 5B). These antimetastatic effects of RF16 were not due to cytotoxicity (Figure 2D).

### 2.4. RF16 Diminished IL-8-Induced EMT of Breast Cancer Cells

To explore the effect of RF16 on the IL-8-induced EMT of breast cancer cells, EMT markers were detected on IL-8-stimulated and RF16-treated MCF-7 and MDA-MB-231 cells. Compared with the control group, IL-8 stimulation and RF16 treatment did not induce any difference in the levels of the ‘epithelial’ marker E-cadherin and the ‘mesenchymal’ marker fibronectin in MCF-7 cells (Figure 6A). However, following IL-8 stimulation, the E-cadherin levels were significantly downregulated by 0.81-fold, whereas the fibronectin levels were significantly upregulated by 2.77-fold in the MDA-MB-231 cells (Figure 6B). Similar results were obtained for migration and invasion, indicating that MDA-MB-231 cells were more sensitive to IL-8 stimulation and that RF-16 attenuated IL-8-induced EMT. Similarly to the protein levels, the mRNA levels of E-cadherin, fibronectin, and MMP2 were upregulated by IL-8, and RF16 treatment reversed the IL-8 function on mRNA expression (Figure 6C). These results indicate that RF16 reduced IL-8-induced EMT in MDA-MD-231 cells.

### 2.5. RF16 Blocked IL-8-Induced PI3K/AKT and p38 MAPK Phosphorylation

IL-8 can promote the EMT-mediated activation of AKT signaling and treatment resistance in breast cancer cells [31]. It also activates MAPK/ERK and p38 signaling, which are crucial modulators of EMT [25,35,36]. Therefore, we determined the activation of AKT, ERK, and p38 in IL-8-stimulated MDA-MD-231 cells treated with RF16. Western blotting revealed significant p-AKT and p-p38 activation in IL-8-treated MDA-MD-231 cells, and RF-16 treatment significantly suppressed p-AKT and p-p38 (Figure 7). However, ERK phosphorylation was unaffected by IL-8 or RF16 treatment. These results indicate that RF16 inhibits EMT by inhibiting the downstream signal p-AKT and p-p38 activation in MDA-MB-231 cells.

### 2.6. RF16 and Docetaxel Synergistically Suppress MDA-MB-231 Cell Growth Xenografted in SCID Mice

The IL-8–CXCR1/2 pathway is involved in the tumorigenesis of different tumor xenograft models [16,37]. We examined whether RF16 inhibits this pathway in breast cancer cells in SCID mouse xenograft models. Compared with the control group, RF16 and docetaxel monotherapy only slightly increased the survival rate in mice, and their combination significantly increased the survival rate (Figure 8A). A comparison of tumor volumes revealed that long-term low-dose weekly docetaxel slightly increased the tumor size, RF16 alone reduced the tumor volume, and their combination reduced the tumor volume by as much as 50% (Figure 8B). RF16 did not affect mouse body weight (Figure 8C) but decreased tumor weight (Figure 8D). In addition, tumors obtained from the mice cotreated with RF16 and docetaxel were significantly smaller than those obtained from the untreated control (Figure 8E). These data support those of in vivo studies indicating that RF16 can inhibit breast cancer cell growth and increase the antitumor efficacy of docetaxel.

## 3. Discussion

IL-8 can modulate the survival and proliferation of solid cancers [38]. By binding to its receptors, CXCR1 and CXCR2, IL-8 supports breast cancer progression by promoting tumor cell invasion and metastases [36,39]. Therefore, inhibition of this binding may be used as a therapeutic target against breast cancer. We investigated the effect of the previously designed peptide RF16 [34], which competes with IL-8′s binding to CXCR1/2, on the growth, proliferation, migration, and invasiveness of MCF-7 and MDA-MB-231 cells.

The WST-1 assay and colony formation assay revealed that RF16 markedly reduced the growth of IL-8-stimulated MDA-MB-231 cells but only slightly influenced IL-8-stimulated MCF-7 cells. Notably, IL-8 expression is higher in ER− breast cancer, which has a poorer prognosis, than in ER+ breast cancer [40]. This may explain why RF16 more efficiently suppressed the ER− MDA-MB-231 cells than the ER+ MCF-7 cells. RF16 dose-dependently suppressed IL-8-stimulated MDA-MB-231 cells, whereas little change was observed when cells were treated with RF16 alone.

EMT plays a vital role in metastasis, which causes tumor cells to migrate and adhere to the ECM while secreting MMPs. After ECM degradation, tumor cells can invade the blood vessels, migrate, and metastasize to distant tissues or organs. We used multiple modalities to detect cancer cell migration, invasion, and EMT. The migration ability of MDA-MB-231 cells was significantly strengthened by IL-8 stimulation and inhibited by RF16. However, in MCF-7 cells, the migration abilities after the IL-8 stimulation and colony formation experiments were similar, supporting our speculation that MCF-7 cells would have extremely low sensitivity and responsiveness to IL-8. Similar results have been obtained in other studies [41,42]. In the cell invasion experiment, the synthetic peptide drug RF16 weakened the invasion ability induced by IL-8. In addition, IL-8 stimulation significantly increased MMP-9 activity, which was decreased with RF16 treatment.

We also detected the expression of upstream EMT marker proteins and mRNAs, such as E-cadherin and fibronectin. E-cadherin is a vital tumor development suppressor gene, and the loss of this protein adhesion can turn benign tumors into aggressive malignant tumors [43,44]. Fibronectin can increase in expression and promote tumor growth, migration, and invasion in cancer. For this reason, fibronectin can be used as a marker of malignant tumors [45,46]. The protein and mRNA expression levels were detected using Western blotting and qRT-PCR. E-cadherin expression decreased and fibronectin expression increased in MDA-MB-231 cells upon stimulation by IL-8. However, RF16 treatment can reverse IL-8-induced EMT expression.

IL-8 stimulates breast cancer cell growth via the PI3K/AKT and MAPK/ERK pathways [35,47]. Our findings indicate that RF16 reduces AKT and p38 phosphorylation in IL-8-stimulated breast cancer cells, which is consistent with the suppressed proliferation after RF16 treatment (Figure 7). Many other signaling pathways, such as NF-κB, TGF-β, and Notch signaling, are involved in breast cancer cell growth and development [48,49,50] and may therefore be involved in or affected by RF16 treatment. In addition, whether breast cancer cells have breast cancer susceptibility gene 1 (BRCA1) and BRCA2 mutations affects their responses to treatment differently [51,52]. Therefore, the next stages of research will involve the identification of the signaling pathways and the relationship between tumor cell mutations and the antitumor effects of RF16.

In the in vitro experiments, RF16 inhibited the development of tumors by reducing the affinity to CXCL8/CXCL1/2, leading to suppressed migration and invasion. However, the inhibitory function of RF16 on IL-8-stimulated responses may not be dose-dependent. The possible reasons include the peptide being too short, being absorbed by cells for other purposes, or being attracted to other peptide drugs at some concentrations. This is why the effects of peptide drugs are unstable.

We used SCID mice for in vivo experiments and injected cancer cells to establish a tumor model and simulate a mouse breast cancer model. RF16 treatment reduced the breast-cancer-associated mouse mortality, and the effect was stronger when combined with low-dose docetaxel. Docetaxel combined with RF16 decreased both tumor weight and size (Figure 8).

Our results indicate that RF16 significantly inhibited IL-8-stimulated cell growth, migration, and invasion in MDA-MB-231 breast cancer cells by blocking the activation of p38 and AKT cascades. In addition, RF16 can potentiate the antitumor effect of docetaxel. These results indicate the high potential of RF16 in cancer treatment in combination with anticancer drugs.

## 4. Materials and Methods

### 4.1. Molecular Docking

The preferable binding of RF16 docked to CXCR1/2 was performed using the Molecular Operating Environment (MOE, http://www.chemcomp.com (accessed on 13 October 2020)) software package (MOE2020.09). The structure of RF16 was truncated from the complex structure of the CXCL8-CXCR2-Gi protein (PDB: 6LFO, [53]). The first solved CXCR1 structure is inactive without any agonist bound (PDB: 2LNL, [54]) and lacks the extracellular N-terminus. Because of the high sequence similarity (76.4%) for CXCR1/2, homology modeling was also performed to construct CXCR1 with a part of the N-terminal region. Finally, RF16 was used to redock to CXCR1/2, and this was compared with the solved CXCL8–CXCR2 complex structure.

### 4.2. Materials

The RF-16 peptide (RCQCIKTYSKPFHPKF) was based on the in silico analysis from our previous study [34]. CF25 was used as the scramble peptide (CPLNGSTVYGHLRHCLSCSGTMVKF). All peptides used in this study were chemically synthesized by GeneMark (GMbiolab, Taipei, Taiwan) with a solid-phase methodology.

### 4.3. Cell Lines and Cell Culture

MCF-7 and MDA-MB231 cells were purchased from the Food Industry Research and Development Institute (Hsinchu, Taiwan). The cells were maintained in Dulbecco’s modification of Eagle’s medium (DMEM; Gibco; Thermo Fisher Scientific, Waltham, MA, USA), supplemented with 2 mM of L-glutamine, 100 U/mL of penicillin (Biowest, Riverside, MO, USA), and 100 mg/mL of streptomycin (Biowest, Riverside, MO, USA), and 10% fetal bovine serum (FBS; Gibco, Waltham, MA, USA). The human microvascular endothelial cell line HMEC-1 (ATCC, No. CRL-10636) was received from American Type Culture Collection (Teddington, UK) and maintained in MCDB-131 medium (Gibco, Waltham, MA, USA) with endothelial cell growth supplement (Millipore, Billerica, MA, USA) and 15% FBS, as previously described [55]. All cells were incubated at 37 °C in a humidified atmosphere of 5% CO_2_.

### 4.4. Cell Viability Assay and Proliferation Assay

To evaluate cell viability, 1 × 10^4^ cells were seeded in 96-well plates. Upon reaching confluence, the cells were added to a medium containing RF16 and CF25 peptides at various concentrations. After 24 or 48 h of incubation, cell viability was determined using the WST-1 assay (Roche, Indianapolis, IN, USA), in accordance with the manufacturer’s instructions. The absorbance was measured at 450 nm.

To evaluate cell proliferation, the cells were seeded at a density of 2 × 10^3^ cells/well in 96-well plates. After 2 h, the cells were treated with different concentrations of RF16 with or without IL-8 (PROSPEC, Ness-Ziona, Israel). The WST-1 assay was performed at 24 and 48 h post treatment. The absorbance was measured at 450 nm.

### 4.5. Colony Formation

First, 1 × 10^3^ cells/well in 2 mL of medium were seeded in 6-well plates. The cells were then cultured at 37 °C in an incubator with 5% CO_2_ for 2 weeks, with the medium changed every 3 days. After colony formation, the cells were washed with PBS, fixed with 75% ethanol, and stained with 0.1% crystal violet. The colonies were treated with 20% acetic acid in methanol, and the absorbance was measured at 540 nm.

### 4.6. Wound-Healing Assay

First, 3 × 10^5^ cells were seeded in 12-well plates for 24 h. Before a scratch was created with the tip of a pipette, the cells were serum-starved overnight. After cell debris was removed through washing with glucose potassium sodium phosphate solution (GKNP), the cells were maintained in a condition medium containing 0.5% FBS and indicated concentrations of RF16 with or without IL-8. The cell culture was photographed at 0 and 24 h by using an Olympus CKX41 phase contrast microscope (Olympus Corporation, Tokyo, Japan) at 100× magnification. The width of the remaining wound area relative to the initial wound width was determined.

### 4.7. Invasion Assay

Migration of breast cancer cells through gelatin-coated filters was performed using the Transwell System (BD Bioscences, San Jose, CA, USA). Transwell (polycarbonate membrane with 8 µm pore size) in a 24-well plate was used. A total of 0.1% gelatin coated on Transwell filters was treated with various concentrations of RF16. Breast cancer cells (2 × 10^5^ cells/well) were added to the upper cavity of the Transwell insert containing 50 μL of DMEM. To initiate migration, 25 ng/mL IL-8 was added to the lower chamber. After 8 h of incubation at 37 °C under 5% CO_2_, the cells transferred to the lower chamber were stained and counted as metastatic breast cancer cells under a microscope. All experiments were performed independently three times.

### 4.8. Gelatin Zymography

The gelatinolytic activity of matrix metalloproteinases (MMPs) in the culture medium was detected using a gelatin zymography assay. Media from the treated cells were subjected to 8% SDS-PAGE containing 0.1% gelatin (Sigma-Aldrich, St. Louis, MO, USA). After gel electrophoresis, the gels were washed twice with washing buffer (2.5% Triton X-100 and 50 mM Tris HCl, pH 7.4) and incubated in incubation buffer (50 mM Tris HCl, pH 8.0; 5 mM CaCl_2_, 1 μM ZnCl_2_, and 0.05% (*w*/*v*) NaN_3_) for 24 h at 37 °C. The gels were then stained with Coomassie Brilliant Blue R-250 (Sigma-Aldrich, St. Louis, MO, USA).

### 4.9. Protein Extraction and Western Blotting

First, 8 × 10^5^ breast cancer cells/well were cultured to confluence in a 6-cm-diameter dish. Next, the cells were treated with the indicated concentrations of RF16 with or without IL-8 for 24 or 48 h. After washing twice with GKNP, proteins were extracted using RIPA Lysis Buffer (Thermo Fisher Scientific, Waltham, MA, USA). Extracted proteins were separated on 8% SDS-PAGE gels and transferred to PVDF membranes. The membranes were blocked with 5% nonfat milk and then incubated with fibronectin, E-cadherin, and α-tubulin antibodies, followed by incubation with HRP-conjugated mouse or rabbit secondary antibodies (EMD Millipore, Burlington, MA, USA) prior to the development of the membranes by using ECL detection reagent (EMD Millipore). Protein expression was quantified with Image J (National Institutes of Health, Bethesda, MD, USA) and expressed relative to that of α-tubulin.

### 4.10. RNA Isolation, cDNA Synthesis and Quantitative Real-Time PCR

Total RNA of breast cancer cells was extracted using TRIzol reagent (Invitrogen; Carlsbad, CA, USA) in accordance with the manufacturer’s protocol. Next, 5 μg of RNA was used for reverse transcription with the RevertAid First Strand cDNA Synthesis Kit (Fermentas International, Burlington, ON, Canada). A quantitative real-time PCR (qRT-PCR) reaction was performed with the LabStar SYBR qPCR Kit (Bioline, London, UK) by using the Roche LightCycler 480 system under the following conditions: 40 cycles of denaturation at 95 °C for 15 s, 60 °C for 20 s, and extension at 72 °C for 15 s. The oligonucleotide primers used were specific for fibronectin (forward, 5′-GGGCTCGCTCTTCTGATTATT-3′ and reverse, 5′-CTGAGACCACCATCACCATTA-3′); E-cadherin (forward, 5′-GGGTGTCGAGGGAAAAATAGG-3′ and reverse, 5′-CGAGAGCTACACGTTCACGG-3′); and β-actin (forward, 5′-CTAAGTCATAGTCCGCCTAGAAGCA-3′ and reverse, 5′-TGGCACCCAGCATGAA-3′). Relative gene expression was calculated using the ΔΔCt method, and gene expression levels were normalized to β-actin control.

### 4.11. Tumor Xenograft Model

First, 1 × 10^6^ MDA-MB-231 cells were orthotopically injected into the mammary fat pads of 10-week-old female SCID mice. When the diameter of the xenograft tumor reached 0.5 cm, the mice were randomly assigned to 4 groups: docetaxel-treated (10 mg/kg/week, intraperitoneal, for 6 weeks), RF16-treated (5 mg/kg/week, intravenous, for 6 weeks), docetaxel- and RF16-cotreated, and normal-saline-treated (intravenous, for 6 weeks) mice (8 per group). Body weight and tumor volume were recorded every 3 days. The tumor volume was measured using the following equation: V = (L × W2) × 0.5, where L and W represent the length and width of the xenograft, respectively. The mice were monitored daily, and tumor volume was measured every 2 days. The mice were killed 36 days after treatment or when the tumor reached 150 mm^3^ (to avoid tumor necrosis), whichever came first. At the end of the experiments, tumors were isolated from the killed mice and weighted.

### 4.12. Statistical Analysis

All statistical analyses were performed using GraphPad Prism 5 software (GraphPad Software, La Jolla, CA, USA). Data are presented as the mean ± standard deviation (SD) from at least three independent experiments. A one-way analysis of variance was used, followed by a Turkey’s post hoc test for comparisons between groups. *p* < 0.05 was considered statistically significant. FDR (False Discovery Rate) adjustment method was performed in all calculation. The q-value was calculated based on the Benjamini–Hochberg method. ^$^ q-value < 0.05 was considered statistically significant.

## Figures and Tables

**Figure 1 ijms-24-14029-f001:**
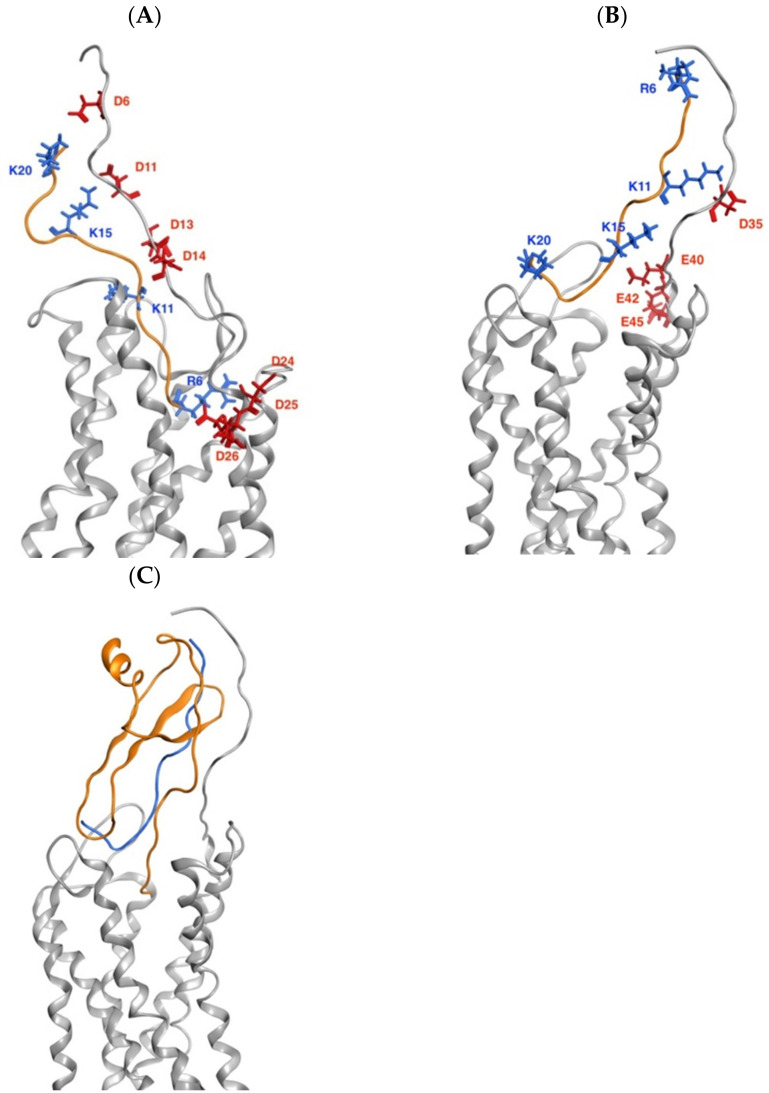
Preferable docking poses of RF16 binding to CXCR1/2 dominated by electrostatic interactions. (**A**) RF16 docking to CXCR1. The orange ribbon is RF16, with positively charged residues labeled in blue. The gray area is CXCR1, with the negatively charged residues labeled in red interacting with RF16. (**B**) RF16 docking to CXCR2. The orange ribbon is RF16, with positively charged residues labeled in blue. The gray area is CXCR2, with the negatively charged residues labeled in red interacting with RF16. (**C**) Superposition of RF16 on the CXCL8-CXCR2 complex structure. The gray area is CXCR2, the orange area is CXCL8, and the blue ribbon is RF16.

**Figure 2 ijms-24-14029-f002:**
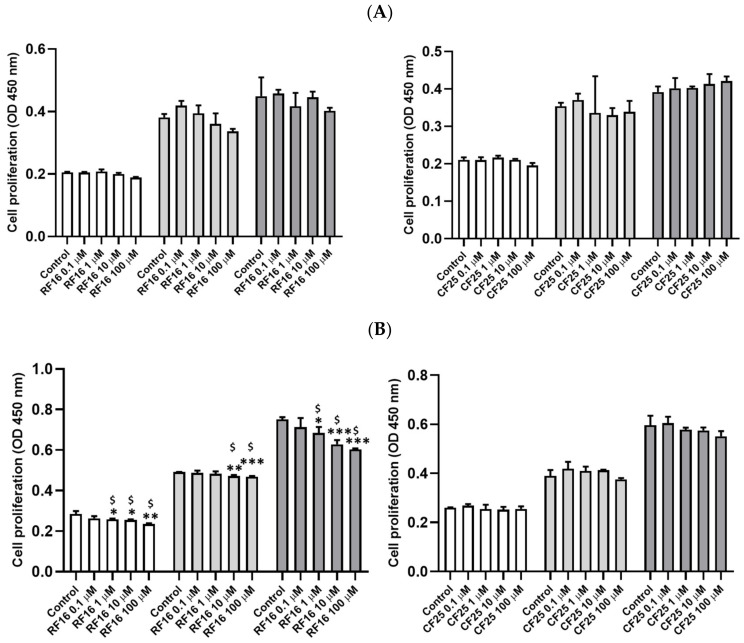
Effect of RF16 and CF25 peptides on cell proliferation and viability in MCF-7 and MDA-MB-231 cells. MCF-7 (**A**,**C**) and MDA-MB-231 cells (**B**,**D**) in a 96-well microplate were treated with various concentrations of RF16 or scramble peptide CF25. After 24 and 48 h of incubation, cell proliferation (**A**,**B**) and viability (**C**,**D**) were evaluated using the colorimetric WST-1 assay. Values are the mean ± SD from three independent experiments. * *p* < 0.05, ** *p* < 0.01 and *** *p* < 0.001 compared with the control group. ^$^ q-value < 0.05 was consider statistically significant.

**Figure 3 ijms-24-14029-f003:**
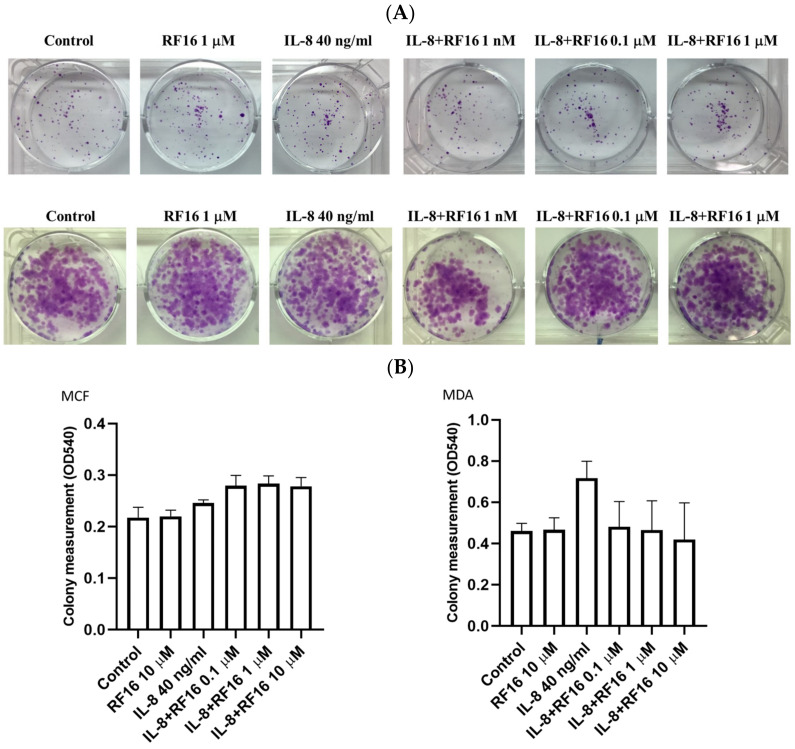
Effect of RF16 on cell growth in MCF-7 (**top**) and MDA-MB-231 cells (**bottom**). (**A**) MCF-7 and MDA-MB-231 cells were incubated with various concentrations of RF16 with IL-8 treatment for 10–12 days. Colony formation was assessed to detect the effect of the RF16 on cell growth. (**B**) Colony formation was assessed and presented as bar graphs. Each column represents the mean ± SD with triplicate experiments.

**Figure 4 ijms-24-14029-f004:**
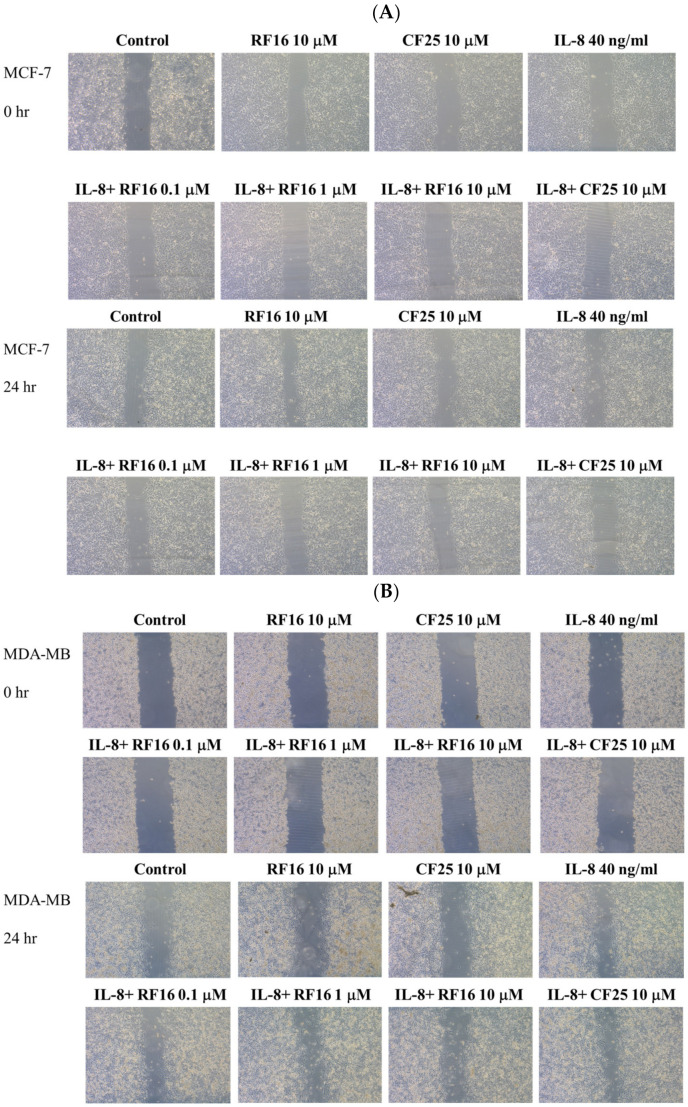
Effect of RF16 on IL-8-induced cell migration in MCF-7 and MDA-MB-231 cells. MCF-7 (**A**) or MDA-MB-231 cells (**B**) were incubated with various concentrations of RF16 with IL-8 treatment. Cell migration was analyzed with scratch wound assay.

**Figure 5 ijms-24-14029-f005:**
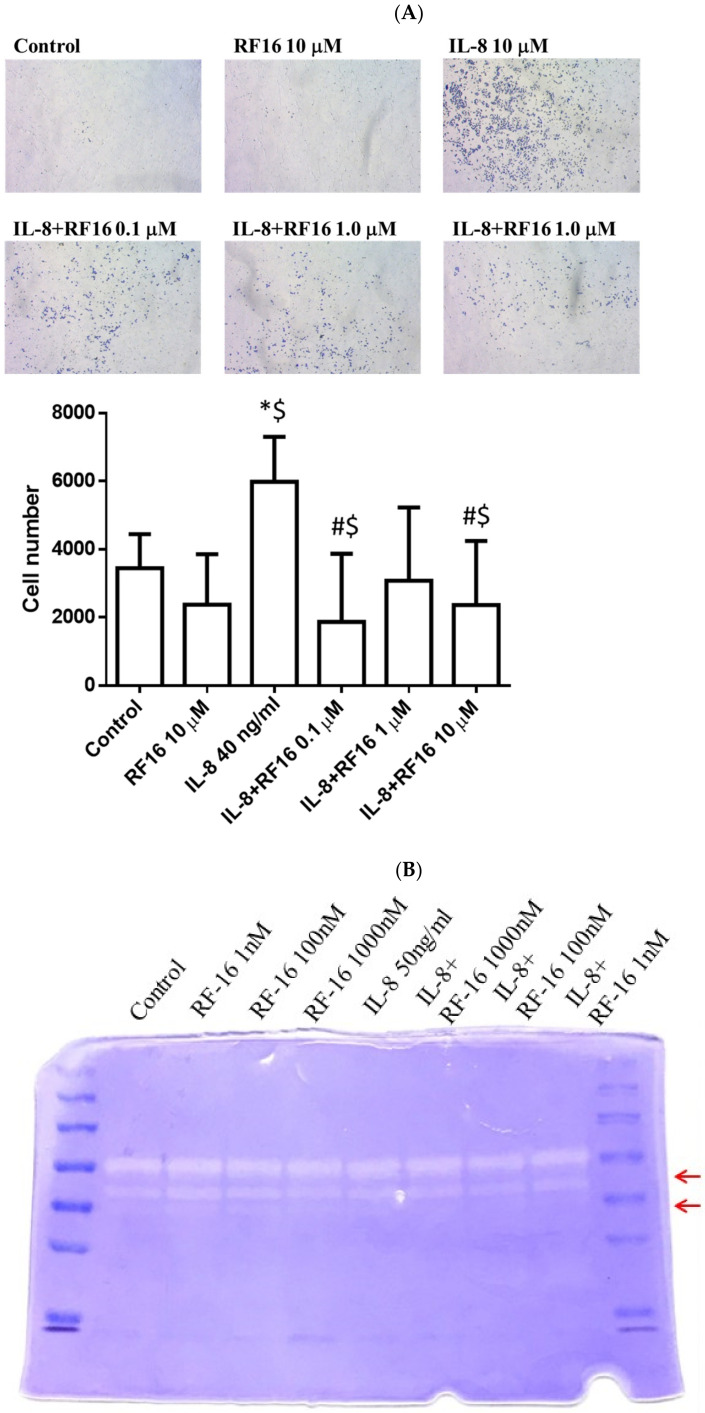
RF16 downregulated IL-8-induced invasion in MDA-MB-231 cells in the in vitro Transwell assay. Cells incubated with IL-8 in the absence or presence of RF16 pretreatment in serum-free DMEM were added to the upper chamber, and medium supplemented with 10% FBS in the absence or presence of IL-8 was placed in the lower chamber. (**A**) Photographs of crystal violet-stained membranes (magnification, 100×). Quantitative analysis of the numbers of migrated MDA-MB-231 cells compared with the control group. IL-8 incubation significantly increased MDA-MB-231 cell migration, which was suppressed by RF16 pretreatment. Values are the mean ± SD from three independent experiments. * *p* < 0.05, versus the control group, and ^#^ *p* < 0.05, versus cells stimulated with IL-8 in the presence of RF16. ^$^ q-value < 0.05 was consider statistically significant. (**B**) RF16 downregulated IL-8 upregulated MMP activity in the gelatin zymography assay. Red arrows indicate the differences of MMPs activities.

**Figure 6 ijms-24-14029-f006:**
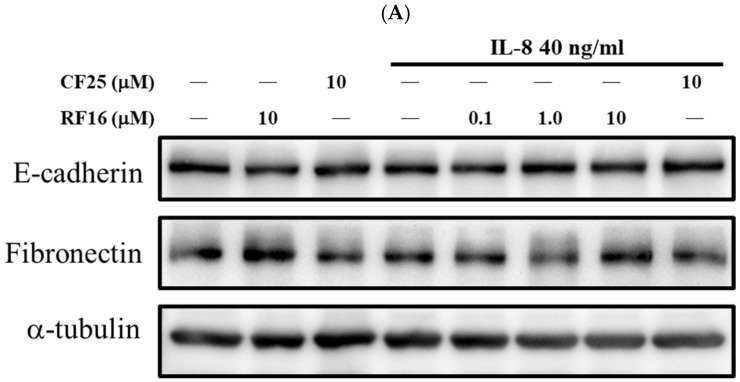
Effect of RF16 on IL-8-induced EMT in cells. MCF-7 (**A**) or MDA-MB-231 cells (**B**) were incubated with various concentrations of RF16 with IL-8 treatment for 24 h. Whole-cell lysates were subjected to Western blot analysis. (**C**) Effect of RF16 on IL-8-induced EMT mRNA transcripts of E-cadherin, fibronectin, and MMP2 in cells. * *p* < 0.05, ** *p* < 0.01 versus control, and # *p* < 0.05, ## *p* < 0.01, and ### *p* < 0.001 versus cells stimulated with IL-8 in the presence of RF16. ^$^ q-value < 0.05 was consider statistically significant.

**Figure 7 ijms-24-14029-f007:**
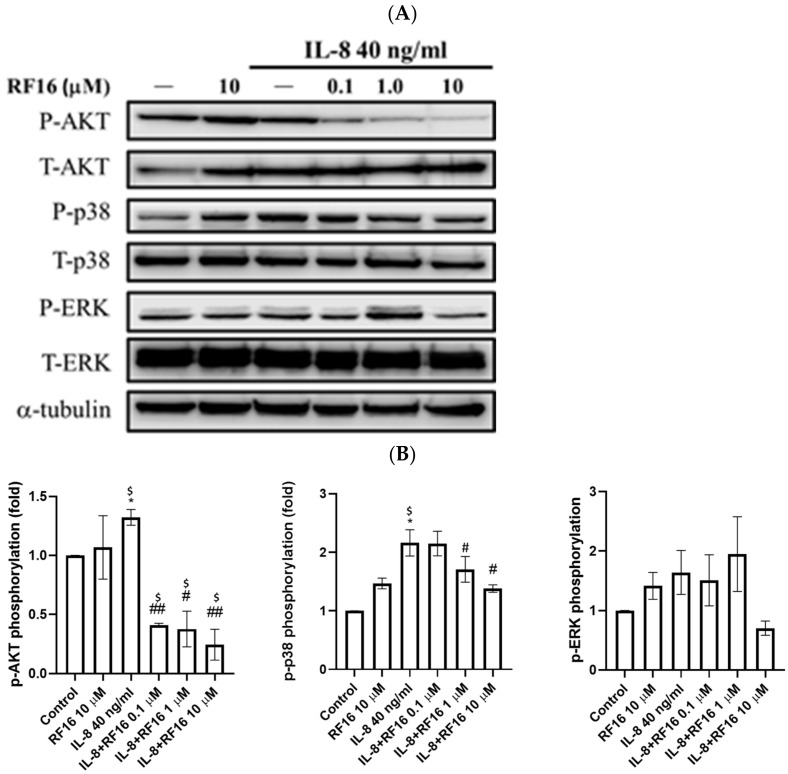
Effect of RF16 on IL-8-increased activation of AKT, ERK, and P38 signaling pathways in MDA-MB-231 cells. (**A**) Western blot indicated that RF16 decreased the IL-8-induced phosphorylation of AKT, ERK, and P38. (**B**) Quantification of AKT, ERK, and P38 activation inhibited by RF16. Values are the mean ± SD from three independent experiments. * *p* < 0.05, versus the control group, and # *p* < 0.05, ## *p* < 0.01 versus cells stimulated with IL-8 in the presence of RF16. ^$^ q-value < 0.05 was consider statistically significant.

**Figure 8 ijms-24-14029-f008:**
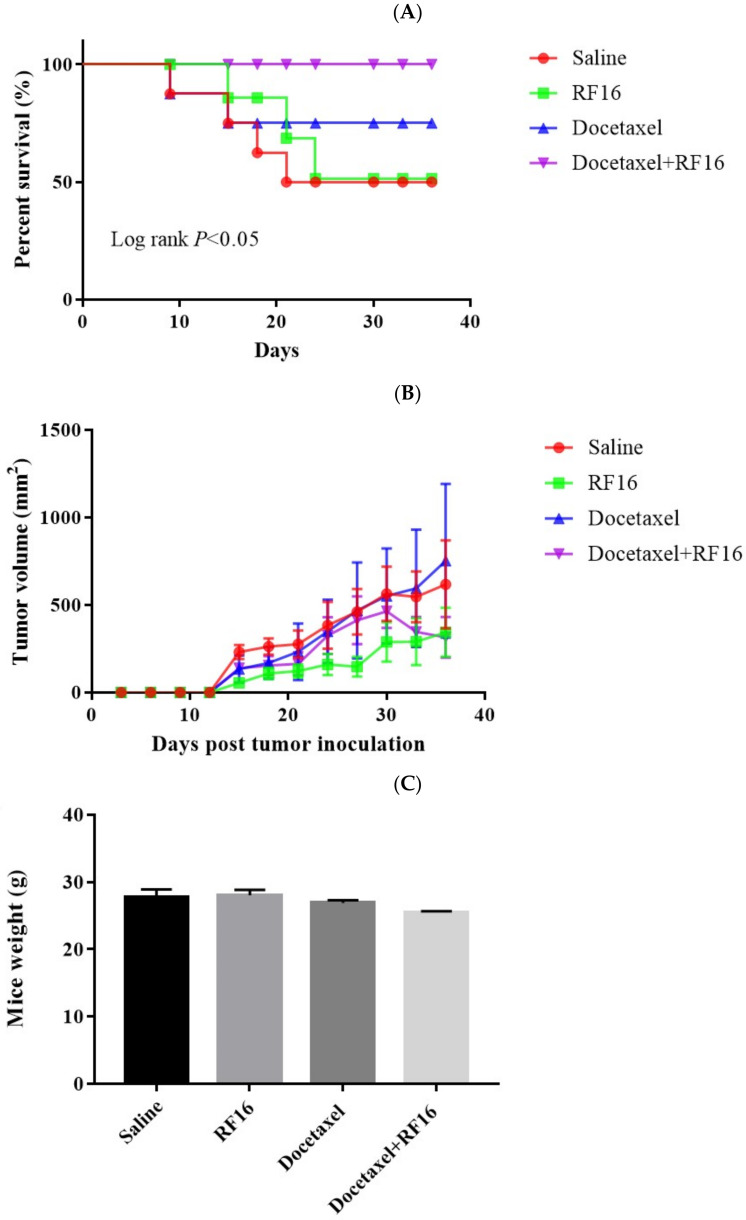
RF16 and low-dose docetaxel synergistically decreased breast cancer growth in a mouse xenograft model. MDA-MB-231 cells (1 × 107 cells) were transplanted to the mammary fat pad of nude mice (eight per group). When the tumor reached a diameter of 0.5 cm, docetaxel (10 mg/kg/week, intraperitoneal, for 6 weeks), RF16 (5 mg/kg/week, intravenous, for 6 weeks), their combination, or normal saline was injected into the mice. (**A**) Mouse survival. (**B**) Tumor volume in the four mouse groups. (**C**) Tumor wet weight at day 36 after transplantation. (**D**) Tumor wet weight at day 36 after transplantation. * *p* < 0.05. (**E**) Photos of tumor xenografts on day 36 after transplantation of human breast cancer cells.

## Data Availability

The data presented in this study are available on request from the corresponding author.

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
