# Peer review of "Therapeutic Peptide RF16 Derived from CXCL8 Inhibits MDA-MB-231 Cell Invasion and Metastasis"

_ijms, 2023, doi:10.3390/ijms241814029_

Round 1

Reviewer 1 Report

Comments and Suggestions for Authors

This is a scientific paper about Therapeutic Peptide RF16, but the quality of the picture and text is somewhat poor. In particular, the figures need to be extensively modified. 

1.English proofreading by an expert is recommended. 

2.It is recommended to delete "based on in silico analysis" from the title. Also, replace or delete the word "Triple Negative Breast Cancer Cell" with MDAMB.

3.Is this the best expression in Figure 1? It needs to reduce the margin and enlarge it to be visible.

4. It is recommended to proceed with the 2D position analysis of Figure 1. If it is difficult, it is better to properly express the surrounding amino acids by greatly expanding the result.

5. It would be good to renumber the pictures as well. See other papers in the journal. Also, the pictures should appear in order in the text. e.g. Figure 2A and then 2C ?

6. Adjust the font size of Figure 5 to be consistent. Figure 6 is the same. Overall, the text in the figures is too small and not clear.

7. The reason for using a peptide called cf25 need to be more clearly explained.

8. Figure 8 should also be modified considering the font size and margins.

9. The Abstract (+ title) should also be rewritten. The logic of that is awkward.

Consider "Our results indicate that RF16 significantly inhibited IL-8-stimulated cell growth, migration, and invasion in MDA-MB-231 breast cancer cells by blocking the activation of p38 and AKT cascades."

10. It would be good to rewrite the Introduction part as well. "Triple negative" and "in silco" are not important in this paper.

11. Why was the cytotoxic drug used at the end (Animal experiment?), and is the EMT story really important? The logic of the overall paper should be considered in depth again when the author rewrite the manuscript.

Comments on the Quality of English Language

English proofreading by an expert is recommended. 

Reviewer 2 Report

Comments and Suggestions for Authors

This study utilized two human breast cancer cell lines, ER-negative MDA-MB-231 and ER-positive MCF-7, to investigate whether RF16 could impede IL-8-induced breast cancer cell growth, proliferation, and invasiveness. Results demonstrated that RF16 effectively reduced cell proliferation, migration, and invasiveness in MDA-MB-231 cells, while also regulating the expression of markers associated with epithelial-mesenchymal transition.

The study is comprehensive and well-presented. However, I have minor concerns:

- The abstract looks short, I suggest adding another 2-3 statements to detail more the methods, result, and the importance of the study.

- the authors may add the false positive rate analysis to adjust and validate the P-values used in the study.

Round 2

Reviewer 1 Report

Comments and Suggestions for Authors

It is recommended to publish after minor revisions.

1.The text in Figure 5, 6, 8 is too small, so it is recommended to enlarge it.

2.It would better to include detailed numbers such as 4.1 and 4.2 in the 4. Materials and Methods section.

Comments on the Quality of English Language

none
